# Agronomic Estimation of Lupin (*Lupinus pilosus* L.) as a Prospective Crop

Oren Shelef [1,*,†] , Eyal Ben-Simchon [1,†] , Marcelo Sternberg [2] and Ofer Cohen [2]

1 Natural Resources Department, Plant Sciences Institute, Agricultural Research Organization (ARO), 68 Maccabim Road, Rishon Le Tzion 7505101, Israel; eyal.ben-simchon@mail.huji.ac.il

2 School of Plant Sciences and Food Security, Faculty of Life Sciences, Tel Aviv University, Tel Aviv 6997801, Israel; marcelos@tauex.tau.ac.il (M.S.); ofercoh77@gmail.com (O.C.)

* Correspondence: shelef@volcani.agri.gov.il; Tel.: +972-526610931

† These authors contributed equally to this work.

**Abstract:** The global dependence on a narrow range of crops poses significant risks to food security, and exploring alternative crops that enhance agrobiodiversity is crucial. *Lupinus pilosus* L., a wild lupin species native to Israel, represents a promising candidate for domestication due to its large seeds and high protein content. This study is the first to evaluate the agronomic potential of *L. pilosus*, focusing on populations from basalt and limestone soils. We hypothesized that *L. pilosus* has significant potential as a novel high-protein crop and that its agronomic characteristics vary among geographically distinct populations. We performed a net-house experiment to test these hypotheses, exploring dozens of agronomic traits for each of the 10 accessions originating in wild populations. We found that basalt-origin accessions exhibited 34.2% higher seed weight, while limestone accessions doubled their seed yield when exposed to honeybee pollination. Notably, high-density cultivation did not reduce seed yield, suggesting that *L. pilosus* could be successfully cultivated under crop-like conditions. Our findings highlight the species' adaptability to different soil types and its responsiveness to pollination, traits that align with the need for climate-resilient crops. This study presents a significant step forward in the domestication of wild lupins, particularly in regions prone to environmental stressors. Compared to other studies on wild lupin domestication, this research provides new insights into the role of ecology in shaping agronomic traits, emphasizing the unique combination of seed yield and plant traits under diverse growing conditions.

**Keywords:** legumes; novel crop; plant ecology; native plants; resilience





## 1. Introduction

### 1.1. Sustainable Food Production and Novel Protein-Rich Crops

It is widely agreed that the global need to produce food more sustainably is increasing in the face of growing demands and agriculture-associated environmental pressures [1,2]. Globally, the majority of crop production in arable lands is cultivated with minimal crop diversity, thus narrowing genetic agrobiodiversity [3,4]. This uniformity facilitates the use of large machines as an efficient way to produce large amounts of food [5]. However, narrowing agrobiodiversity exposes food systems to risks due to reduced crop resilience, as increased homogeneity leads to increased vulnerability [6,7]. Furthermore, the loss of biodiversity severely limits our ability to develop new cultivars, constraining future crop diversification [8]. Ecological homogeneity is linked to agricultural challenges such as foreign species invasions, pest outbreaks, and susceptibility to pathogens or climate changes [6], and ultimately, yield loss [9]. Historically, this has led to catastrophic failures and significant famine events like the Irish Potato Famine, which resulted from the vulnerability of a single potato clone to disease [10]. Similarly, today, global banana production is threatened by Panama disease [11], stressing how a drastic narrowing of crop diversity is threatening global crops and food security.

To address these challenges, agricultural researchers and practitioners are exploring innovative strategies to diversify monocultures and enhance agroecosystem resilience [12,13]. In addition, intra-cultivar diversity [14] may significantly impact the function of an agroecosystem. For example, cover crops improve water balance and soil health [15], mixed cropping affects microbial soil diversity [16], and alley cropping can enhance arthropod diversity [17]. Native plants are likely to better tolerate the local environment [18–21], supporting regenerative agriculture. The use of wild plants has vast and almost unexplored potential to provide more sustainable food chains [21–23]. Furthermore, certain crops, such as lupins, offer economic advantages, with significantly lower farming costs compared to other legumes [24]. Diversifying local plant-based protein production will help to foster more sustainable agroecosystems, create more efficient supply chains, and ultimately increase food security. *L. pilosus*, a local legume in Israel, has evolved to fit the local environmental conditions over time, offering potential benefits for sustainable agriculture in the region.

Beyond agricultural diversification, addressing impacts on the human diet is another essential aspect of enhancing food system sustainability. Accumulated evidence repeatedly shows that meat and dairy production has a significantly greater environmental footprint than plant crops [25]. The awareness of these environmental consequences has propelled the annual growth of the meat substitute industry [26] to between 5 and 20% in Europe [27]. Equipped with the ability to accurately manipulate target genes for de novo domestication [28], as well as technical methods to determine the nutritional value and solve potential dietary or health risks, developing novel protein crops is more accessible now than ever. Here, we examined the agronomic aspects of recruiting wild lupin populations as a novel protein crop.

### 1.2. Lupins as a Protein Crop

Lupins (*Lupinus* spp.) are known for their large seeds and high-value proteins [29]. Four species (*L. albus* L., *L. angustifolius* L., *L. luteus* L., and *L. mutabilis* Sweet) are globally cultivated and used in food markets [30], and their role as novel proteins are regularly studied [31,32]. While these species have been well-studied, there is growing interest in exploring the potential of other lupin species, particularly *L. pilosus*, as novel protein crops [33,34]. Other lupins have never been domesticated as a commercial food source due to their high alkaloid content, which makes their seeds unsuited for human consumption before debittering [35]. Studies on *L. pilosus* have shown promising results in terms of protein content and adaptability to various environmental conditions [36,37]. Interestingly, *L. pilosus* showed a good potential as a crop for a coffee substitute [38].

There are four main components necessary for determining the success of wild lupins as a novel plant-based protein resource: protein content, alkaloid content, seed yield and agricultural compatibility. Agricultural compatibility includes secondary benefits such as the nitrogen enrichment of soils for subsequent crops [39]. Studies have shown that environmental conditions can affect the profile of alkaloids, metabolism of amino acids, and agronomic aspects (such as plant phenology, morphology, and tolerance to various soil conditions [40,41]).

### 1.3. L. pilosus as a Potential Crop

Studies conducted in Western Australia, the Department of Primary Industries and Regional Development (DPIRD), investigated lupin species as crops, including rough seeded lupins such as *L. pilosus* [42]. The researchers studied 56 accessions of *L. atlanticus*, and 90 accessions of *L. pilosus*, and they draw the following conclusions regarding the potential domestication of *L. pilosus* [43]: (1) *L. pilosus* phenology is similar to *L. albus*, with early blooming (75–90 days from sowing) and slow pod maturity; (2) *L. pilosus* grows well in a variety of conditions, but suits areas with long seasons for the long maturity of big seeds in a big thick pod; (3) *L. pilosus* seeds have lower protein levels (26% in average) in comparison to the soft-seeded lupins such as *L. albus* (39% protein) and *L. angustifolius* (34%

protein). This may be explained by the thickness of the seed coat and pods. Researchers [43] used the cross-breeding of artificial and wild mutations to target domestication traits, such as non-shattering, high yield, and sweetness—low alkaloid content [35,44].

*L. pilosus*, like many lupin species, is predominantly self-pollinated but retains the capacity for facultative cross-pollination [45,46]. While primarily self-pollinated, *L. pilosus* attracts insect visitors, including bees, with its colorful flowers and nutritious pollen. Despite lacking nectar, it offers pollen as a reward. The species exhibits a floral color change from white to purple, which may enhance pollinator attraction and potentially promote outcrossing [46].

Our investigation is focusing on the agronomic production potential of *L. pilosus* by examining wild populations from diverse habitats across Israel. We evaluate the agronomic properties of wild *L. pilosus* accessions grown under net-house conditions. This approach establishes a foundational understanding for the long-term process of lupin breeding and domestication adapted specifically for local conditions. *L. pilosus* is an exciting potential novel legume crop for the following reasons: (1) *L. pilosus* produces large seeds in natural populations (~550 mg per a single seed); (2) *L. pilosus* tolerates a relatively wide range of alkaline and acidic soils, including lightly alkaline soil with pH > 7; (3) lupins are among the most protein-rich legumes [47]. To identify focal domestication traits for this research, we looked at the literature related to lupin agronomy [48] and de novo domestication [28], and the references therein. Table 1 summarizes the agronomic traits that we covered for this research.

The objectives of this study are as follows:

(1) Provide an initial estimation of the potential yield of seeds per cultivated area for *L. pilosus* in common garden net-house conditions;
(2) Compare agronomic traits of different accessions aiming to link environmental factors (climatic, edaphic) to agronomic proprieties;
(3) Study the effects of pollination on seed production in accessions originating from wild *L. pilosus* accessions;
(4) Evaluate two plant growing densities, low density to simulate optimal growth conditions, and high density to mimic agricultural crop conditions.

First, we hypothesized that *L. pilosus* has potential to serve as a novel high-protein crop, showing high agronomic performance traits. Second, we hypothesized that the geographically distinct populations of *L. pilosus* possess different agronomic characteristics. To examine our hypotheses, we collected seeds from 10 wild populations, let them grow in a net-house experiment, and evaluated agronomic parameters throughout the growth period.

**Table 1.** List of the examined domestication agronomic traits.

| Type | Trait | Value | Method |
|---|---|---|---|
| Vegetative phenology | Height growth | cm | meter |
| | Width growth | cm | meter |
| | Vegetative development | Index rank | rank |
| Architecture | Leaf number development | # | count |
| | Branch number | # | count |
| Performance | Chlorosis | Index rank | count |
| | Lodging | # | count |
| | Photochemical efficiency | Fv/Fm | miniPAM |
| | Pathology | # | count |
| | Leaves DW | g | weight |
| | Stem and branch DW | g | weight |
| | Vegetative biomass DW | g | weight |

**Table 1.** *Cont.*

| Type | Trait | Value | Method |
|---|---|---|---|
| Reproductive phenology | Bloom development | Index rank | rank |
| | Bloom structure | Index rank | rank |
| | Lateral pods development | Lateral bloom branch # | count |
| | Main stem pod development | # | count |
| | First dry pod | Day from start | observe |
| | Pod ripening time | Day from start | observe |
| | All pods opened | Day from start | observe |
| Yield | Normal seeds number | # | count |
| | Infested seeds number | # | count |
| | Abnormal seeds number | # | count |
| | Number of pods | # | count |
| | Normal seeds in closed pods | # | count |
| | Abnormal seeds in closed pods | # | count |
| | Normal seeds DW | g | weight |
| | Abnormal seeds DW | g | weight |
| | Infested seeds DW | g | weight |
| | Pod valves DW | g | weight |
| | Total ripe fruit number | # | calculated |
| | Fully developed seed number | # | calculated |
| | Fully developed seeds DW | g | weight |
| | Total seed number | # | calculated |
| | Total seed DW | g | weight |
| | Normal seeds rate (%#) | %#, %DW | calculated |
| | Abnormal seeds rate (%#) | %#, %DW | calculated |
| | Infested seeds rate (%#) | %#, %DW | calculated |
| | Developed seeds rate (%#) | %#, %DW | calculated |
| | **Reproductive/Vegetative** | Ratio | calculated |

## 2. Materials and Methods

### 2.1. Lupinus pilosus Accessions

Ten wild populations of *L. pilosus* were collected in natural areas across Israel, covering a range of environmental conditions and two soil types, basaltic protogrumols (basaltic bedrock) and rendzina soils (limestone bedrock) (Table 2). These populations were identified and marked in early spring (February to early March) at the start of the flowering period from 2017 to 2019. In late spring (April to May), matured lupin pods were collected before seed dispersal. Approximately 100 individuals were sampled at each site. The matured pods, still on the plants, were gathered into paper bags and transported to Tel Aviv University. In the lab, pods from each population were placed in separate containers and allowed to dry at room temperature until they opened naturally. Once dried, the lupin pods were removed, and seeds from each population were stored in dry, cool conditions. Seeds from each population were used for the net-house experiment.

**Table 2.** Accession origin and details of plant population. # = accession number; collected = the year in which seeds were collected from the wild population, in the early summer (May–June); site = domestic name of the site location; soil = basalt or limestone; altitude = in meters above sea level (m a.s.l.); location = GPS coordinates, latitude (N) and longitude (E); average seed DW = an average seed dry weight in grams, given as an average weight of a single seed.

| # | Collected | Site | Soil | Altitude (m) | Location | Average Seed DW (g) |
|---|---|---|---|---|---|---|
| 1 | 2017 | Golan—Mapalim | Basalt | 520 | 32°59′10.5″ N 35°45′04.1″ E | 0.498 |
| 2 | 2018 | Hula—Hamdal | Basalt | 100 | 33°05′39.3″ N 35°40′27.0″ E | 0.633 |

**Table 2.** *Cont.*

| # | Collected | Site | Soil | Altitude (m) | Location | Average Seed DW (g) |
|---|---|---|---|---|---|---|
| 3 | 2019 | Judean Mts.—Sokho | Limestone | 330 | 31°40′52.8″ N 34°58′34.4″ E | 0.522 |
| 4 | 2018 | Golan—Ofir | Basalt | 250 | 32°48′53.6″ N 35°39′45.6″ E | 0.863 |
| 5 | 2017 | Golan—Hisfin | Basalt | 420 | 32°50′48.2″ N 35°46′46.4″ E | 0.381 |
| 8 | 2019 | Judean Mts.—Bekoa | Limestone | 150 | 31°49′21.5″ N 34°56′27.6″ E | 0.692 |
| 10 | 2017 | Golan—Hazeka | Basalt | 980 | 33°03′42.7″ N 35°50′47.1″ E | 0.442 |
| 11 | 2017 | Judean Mts.—Matta | Limestone | 620 | 31°42′58.1″ N 35°03′01.1″ E | 0.548 |
| 21 | 2019 | Carmel—Makura | Limestone | 90 | 32°38′22.2″ N 34°59′19.9″ E | 0.622 |
| 26 | 2019 | Samaria—Kedumim | Limestone | 360 | 32°12′53.9″ N 35°09′50.5″ E | 0.558 |

### 2.2. Experimental Design and Plant Growth

We performed a net-house growth experiment (Supplementary Material S1) at the Agricultural Research Organization, Rishon Le Zion, Israel (31°59′ N 34°49′ E). The research site is characterized by a Mediterranean climate with an annual 540 mm average rain precipitation and an average annual temperature of 20 °C, with a typical warm dry summer and a mildly cold wet winter. We categorized the soil as loamy sand (86% sand, 8.9% clay, and 5.1% silt). The growth period began after treating the seeds (physical scarification by sandpaper and 24 h soaking in water) and seeding on 19 November 2019, at the beginning of the wet winter. Plants completed a maximum of 212 days of growth to seed maturation on 17 June 2020. For the establishment period, seeds intended for chambers A–D were first sown in 0.5 L pots. The local soil was inoculated with transported soil from the vicinity of the wild lupin populations to ensure the suitable rhizobia presence for symbiotic N fixation. After a month of establishment in the pots, the plants were transplanted into the soil. Lupins in chamber E were sown directly in the soil without a pot establishment period. Plants were not irrigated except for daily wetting during the establishment period, and the two irrigation cycles, 2 h per cycle, during the first two weeks of growth in soil. The irrigation was conducted by drip irrigation pipe, with controlled emitters along the pipe, located every 30 cm, each emitting 2 L per hour. Temperature and humidity were monitored in each chamber throughout the experiment to ensure the microclimatic conditions were not significantly different between chambers. For the same reason, soil composite samples were taken from each section for nutrient analysis. Soil pH was measured by saturated soil water extraction using a standard pH meter following standard method (SM) 4500 H-B, soil P was measured by using the Olsen extraction method [49], followed by colorimetric analysis based on SM 4500-P-E, soil K tested by saturated soil water extraction followed by flame photometer analysis based on SM 3500-K-B, soil $NO_3$ and $NH_4$ tested by saturated soil water extraction, respectively, followed by UV spectroscopy based on SM 4500-$NO_3$-B and the Macro-Kjeldale method based on SM 4500-$NH_3$-C. Although *L. pilosus* is mainly self-pollinated, honeybee pollination was introduced to potentially enhance yield through cross-pollination, which can increase seed set in legume species. This approach is based on research showing that even predominantly self-pollinating species can benefit from insect-mediated pollination [50]. To prevent unintended cross-pollination and its effect on seed genotype and phenotype, we isolated each lupin population to separate chambers within a net-house. Ten populations were randomly distributed across a 3-block setup (chambers A, C, and D); 10 plants per accession were grown in 10 rows, separated by 60 cm gaps between plants and 70 cm between rows. The spatial distribution of accessions in each block was decided randomly. An additional chamber (B) was designed similarly to

the first three (A, C, and D) but was designated for open pollination by honeybees, which were introduced when lupins began to bloom. The external wall of the pollination chamber (B) was open to allow free foraging by the bees, as honeybees do not tolerate closed rooms. All ten populations were grown in Chamber E, where the high density resembled dense crop conditions (6.25 plants per m², 40 cm × 40 cm gap between plants, compared to 2.38 plants per m², 60 cm × 70 cm gaps between plants). To avoid uncontrolled pod-shattering after fruit ripening, we covered each plant with net bags (see research setup and photos, Supplementary Material S1 and S2).

### 2.3. Vegetative Growth Parameters

Plant dimensions (height and diameter) were measured per individual plant. We measured the plant dimensions periodically—three times during the growth period—at the early vegetative growth stage (59 days from seeding), after first inflorescence of the limestone l accessions (90 days from seeding), and after the complete fruit setting of all accessions (155 days from seeding). We also tracked the number of leaves during the early stages of growth (up to 48 days from seeding) and the number of side branches.

Plant growth stages were reported per individual, using a phenology index of *L. pilosus*. The index includes the following levels of maturation: (1) rosette up to 5 cm; (2) rosette above 5 cm; (3) main stem inflorescence size is at least 1 cm; (4) main stem inflorescence size larger than 5 cm; (5) main stem flowering, at least three fully developed flowers in the lower floor; (6) main stem mid-flowering; (7) main stem flowering end; (8) main stem post-flowering; (9) plant finished blossoming; (10) first ripe pod; (11) all pods ripe. See Supplementary Material S1 for a detailed growth index. We periodically traced the phenological index at 59, 90, 108, 119, 141, 155, and 170 days after seeding. Additionally, we included 17 collection points based on other vegetative measurements representing the plants' phenological development. In total, 24 phenological tracking points were collected, and 75% were between 160 and 200 days after sowing.

### 2.4. Plant Performance

We measured several aspects of the plant performance over the entire development period in the net-house. Photochemical efficiency expressed by the ratio Fv/Fm indicates the proportion of the maximum possible photochemical activity. In healthy plants, this efficiency is about 80% or 0.8 Fv/Fm [51,52]. We measured the darkened leaves of five plants from each accession with a MINI-PAM (Walz GmbH, Effeltrich, Germany) 48 and 59 days after seeding. We measured the dry biomass with an analytical scale following oven drying (75 °C, 48 h). We divided plant shoots into leaves, stems, branches, and reproductive tissues at harvest time. At the early stages of development (36, 48, and 59 days from seeding), we ranked each individual according to a visual chlorosis score with 0—green and healthy; 5—yellow, chlorotic [53]. Throughout the experiment, we recorded agro-physical vulnerability (lodging damage), die-outs, and pathology.

### 2.5. Reproductive Growth Parameters

We used the same phenology index (Supplementary Material S3) in continuation to the vegetative growth phenology levels. The reproductive growth parameters included the bloom development phase, bloom structure, central stem pod development, lateral pod development, and pod ripening period from the first day to the day when all pods were open.

### 2.6. Yield Parameters

We counted and measured the following parameters per individual—the number of typically developed seeds. We let the regular seeds dry in paper bags at room temperature for 60 days to allow nondestructive dehydration. All other plant materials were dried in an oven (75 °C, 48 h). *Infested seeds* were typically developed seeds infested by insects during maturation. We identified the major pest as the beetle *Bruchidius serraticornis*. *Abnormal*

*seeds* were sorted as seeds that did not complete their maturation process due to plant dehydration or disease. We counted and measured the total *number and dry biomass of pods* per individual plant and normal and abnormal seeds in closed pods. The measurements mentioned above allowed the calculation of the total ripe fruit number, average seed weight, average regular seeds per pod, standard seed rate (per count and weight), and the reproductive/vegetative tissue ratio.

### 2.7. Data Analysis

All data analyses used R version 4.0.2 (Team, 2020).

### 2.8. Phenology

To visualize the phenological development of each accession we used the *stat_smooth* function of the *ggplot2* package [54] with local polynomial regression fitting, losing. The yield of *L. pilosus* in net-house conditions and the effect of honeybee pollination on *L. pilosus* performance were analyzed. We first checked the assumptions of normality and homogeneity of residual variances for all the tested variables by visually plotting sample quantiles against theoretical quantiles and residuals against fitted values. Subsequently, we performed Shapiro–Wilk and Levene's tests for variables that did not meet the above assumptions, and square root and log transformations were used for the analysis. We constructed a linear mix effect model for those variables using the *Elmer* function of the *lme4* package [55] to consider the block effect. Subsequently, we used the ANOVA test followed by the Tukey HSD post hoc test to assess differences between treatments using the *ANOVA* function and the *glut* function of the *multcomp* package [56]. However, if the transformations did not consider normal distribution and homogeneity, we treated those variables with nonparametric methods. For those nonparametric variables, we performed the Wilcoxon test to assess differences between the two treatments by using the *stat_compare_means* function of the *ggpubr* package [57]. In addition, we performed the Kruskal–Wallis test followed by the Dunn post hoc test to assess differences between more than two treatments using the *kruskal.test* function and *dunn.test* function, respectively, in the *FSA* package [58]. For all statistical tests, we considered differences as statistically significant at $p \le 0.05$, with $p \le 0.01$ indicating highly significant differences.

### 2.9. Multi-Dimension Trait Comparison

We reduced the initial dataset to include only the relevant numerical variables for principal component analysis (PCA) and conducted an analysis based on the 18 selected variables. For this analysis, we used the *prcomp* function of the *ggfortify* package [59] and visualized it using the *ggplot2* package [54].

## 3. Results

During the experiment, minimum/maximum temperature, humidity and soil conditions in the net-house were monitored during four separate periods of a few days each. We did not notice substantial differences between the different chambers of the net-house. Generally, the minimum and maximum monitored temperatures were $10.9 \pm 0.7\,^{\circ}C$ and $40.5 \pm 4.8\,^{\circ}C$, respectively, and the minimum and maximum humidity levels were $22.8 \pm 3.3\%$ and $93.6 \pm 3.2\%$, respectively. At the beginning of the growing season, the soil pH was 7.9, and the nutrient levels were 24.9 mg/kg P, 0.37 mEq/L K, 8.4 mg/kg $NO_3$, and 8.1 mg/kg $NH_4$. The nutrient levels were sufficient for growing lupines without the need for additional fertilizers.

Plants from all studied accessions completed their life cycle and produced seeds, showing that *L. pilosus* growth was successful in experimental conditions regardless of accession origin. Initially, we used ten plants per population in low-density conditions (block A, see Supplementary Material S2) and 40 plants per accession in high-density conditions (block E). An average and standard error (SE) of $9.63 \pm 0.09$ out of 10 plants completed their life cycle from each accession in A-D blocks, and of these, an average of

98 ± 1% (SE) produced seeds. The reasons for failure to produce healthy seeds included poor germination and plant disease (2% of all plants). The average maximum plant height and width were 88.5 ± 0.9 cm (SE) and 92.5 ± 1.2 cm (SE), respectively, and the average shoot DW at harvest was 158.5 ± 5.8 g (SE). An average of 13.3 ± 3% (SE) plants per accession suffered chlorosis, but most of the affected plants completed their life cycle despite the apparent chlorosis during the vegetative growth period. *L. pilosus* plants showed high levels of photochemical efficiency (Fv/Fm) both at 48 and 59 days after sowing, with a total average of 0.778 ± 0.02 (SE) and 0.837 ± 0.004 (SE), respectively. The average total seed yield was 2070 ± 246 (SE) kg/ha in the low plant density treatments (blocks A–D) and 198.5 ± 5.9 (SE) seeds per plant, with an average reproductive/vegetative tissue ratio of 1.22 ± 0.05 (SE).

### 3.1. Phenology and Architecture

The entire lifecycle development of *L. pilosus*, from sowing to seed ripening, lasted approximately 200 days (Figure 1a). We observed three phenological stages as follows: vegetative (0–100 days from sowing), inflorescence and fruit setting (50–180 days), and finally ripening (180–212 days). Figure 1a reflects how the accessions differed from each other in the timing period of each stage. The most prominent difference appears in the vegetative to reproductive stage. In accessions 3, 8, 11, 21, and 26, originating from a limestone environment, this developmental transformation occurred a few weeks before it occurred in accessions 1, 5, and 10, originating from a basalt origin. Accession 2, with a basalt origin, showed a phenological growth pattern resembling the limestone accessions, and accession 4, with a basalt origin, showed an intermediate pattern. The short vegetative stage of the limestone accessions was followed by an extended period of inflorescence and fruit setting and, finally, a slightly shorter ripening time than the basalt accessions. We examined the duration from the onset of pod drying to seed shattering. Unlike other phenological differences observed between basalt-origin and limestone-origin accessions, this period did not vary between the two origins or among any accessions, remaining constant at 25.87 ± 0.3 (SE) days.

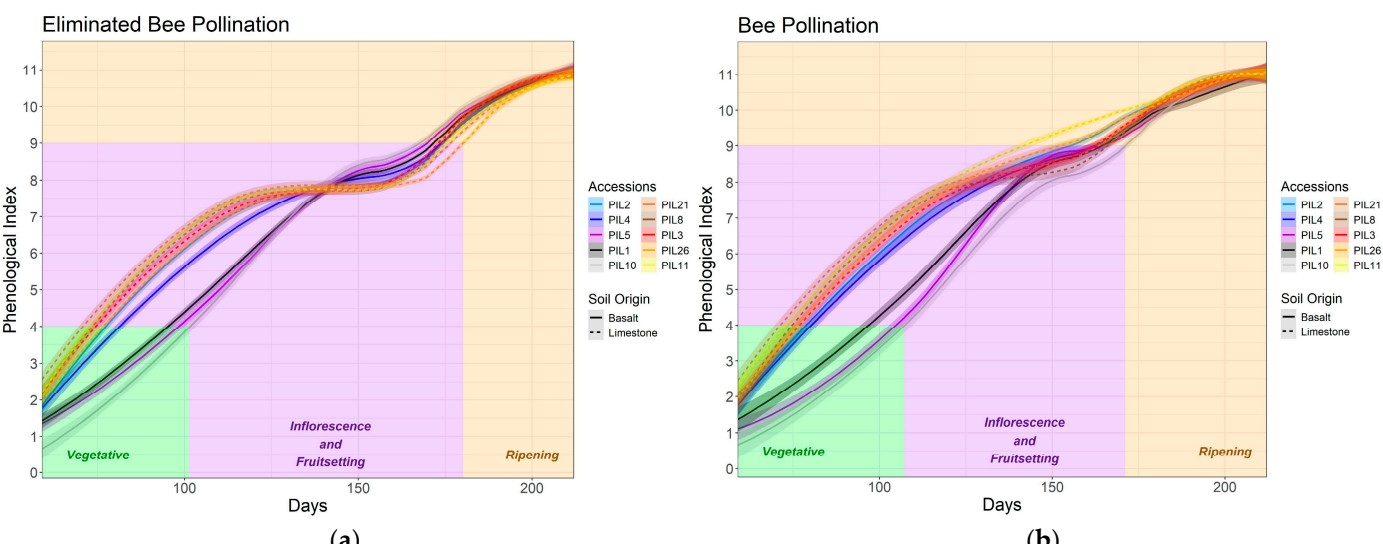

**Figure 1.** Phenology of *L. pilosus* accessions over time. Days are counted from sowing on 19 November 2019 (1) to the completion of the life cycle in all accessions on 17 June 2020 (212). A different background color marks major developmental phases—vegetative phase (green), inflorescence, fruit setting (lilac purple), and ripening (pale brown); (**a**) honeybee pollination excluded; (**b**) with honeybee pollination.

*L. pilosus* plants exposed to spontaneous honeybee pollination showed shorter life cycles, with faster transformations from stage to stage (Figure 2b). The difference in

phenological development between accessions showed a similar pattern when plants were exposed to honeybee pollination (Figure 2b) or when pollination was excluded (Figure 2a).

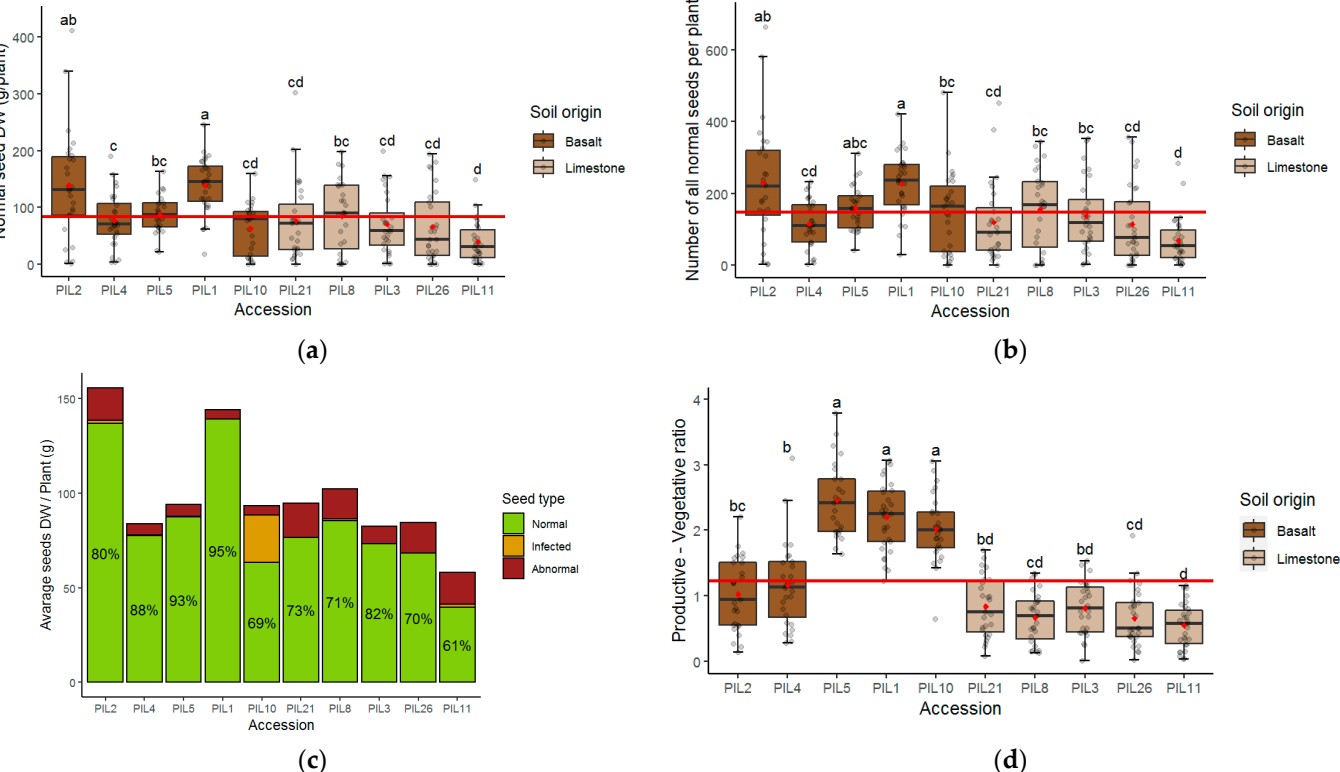

**Figure 2.** Seed yield across the different *L. pilosus* accessions. Accession origin is marked as basalt or limestone, and the population code is listed on the *x*-axis. Statistical differences between accessions, denoted by different letters, are determined by the Kruskal–Wallis nonparametric test followed by the post hoc Dunn test (Benjamani–Hochberg method). In panels a, b, and d, the red dots within the boxplot represent the mean values for each accession, while the red horizontal line indicates the overall mean across all samples. (**a**) Dry weight (DW) of fully developed seeds per *L. pilosus* individual plant by accessions and soil origin. (**b**) Fully developed seed number per *L. pilosus* individual plant by accessions. (**c**) Seed maturation success per population in average seed g DW per plant. Regular seeds—mature intact seeds—appear healthy and viable (percentage of total seeds is denoted); infested seeds are mature seeds with signs of pest damage, presumably by *Bruchidius serrations*; abnormal seeds did not complete their maturation. (**d**) Reproductive/vegetative tissue ratio per accession of *L. pilosus* plants. We measured plant DW at maturation after harvest, separating plants into vegetative and reproductive organs, and calculated the ratio accordingly. The data are normally distributed following a square root transformation. Shapiro–Wilk normality test (W = 0.99, *p*-value = 0.04). Statistical differences between accessions, denoted by different letters, are determined by the ANOVA test followed by the Tukey HSD post hoc test.

## 3.2. Yield of L. pilosus in Net-House Conditions

Normally developed seed weight per plant was significantly higher in the basalt origin accessions, with a $102 \pm 5.6$ (SE) g DW, 34.2% higher as compared to a $67.1 \pm 4.9$ (SE) average in the limestone accessions (Figure 2a). Accessions 1 and 2 showed the highest average seed yield $136.9 \pm 7.9$ g DW (SE) per plant with $221.8 \pm 13.3$ seeds (SE) per plant, while accession 11 exhibited the smallest yield $50.3 \pm 6.1$ g (SE) DW with a seed number of $92.4 \pm 12$ (SE) (Figure 2b). Eventually, the average biomass of a single *L. pilosus* seed was $0.59 \pm 0.01$ (SE) g DW with a maximum of $0.64 \pm 0.01$ (SE) g DW per seed in accession 21, and a minimum of $0.41 \pm 0.02$ (SE) g DW per seed in accession 10.

### 3.3. Seed Development Success Rates

To estimate seed health, we analyzed three different types of seeds per accession (Figure 2c)—fully developed seeds ("normal"), fully developed seeds infested after maturation ("infested"), and seeds that did not develop properly ("abnormal"). Results show that most of the seed DW yield, at least 60–95%, was gained by normally developed seeds in all accessions. Interestingly, the population of 10 showed a large proportion of infested seeds, $23 \pm 6\%$ (SE). The reproductive/vegetative tissue ratio per accession was 253% higher in the basalt accessions (Figure 2d). Accessions 1, 5, and 10 showed significantly higher reproductive/vegetative tissue ratios than all other accessions.

### 3.4. Results of Multi-Dimension Trait Comparison

To explore multi-dimensional differences between the accessions, we performed a PCA of plant traits across accessions (Figure 3). We performed the PCA for 18 factors, including phenological components and aspects of seed yield. The first two principal components explained 56.63% of the sample variance across those variables. The PCA shows that basalt origin accessions (1, 4, 5, and 10) were more distinguished, as their cloud of multi-trait circles is denser. Limestone-origin accessions appear to represent a broader diversity between plants within each accession, with a wider point cloud (3, 8, 11, 21, and 26). Note that in the PCA, accession 2, which originated in basalt soil, exhibits a spread pattern more typical of the limestone accessions. Apart from accession 2, basalt accessions show a multi-dimensional similarity among accessions, focused at the lower left side of the PCA figure, with some minor overlap with the limestone accession.

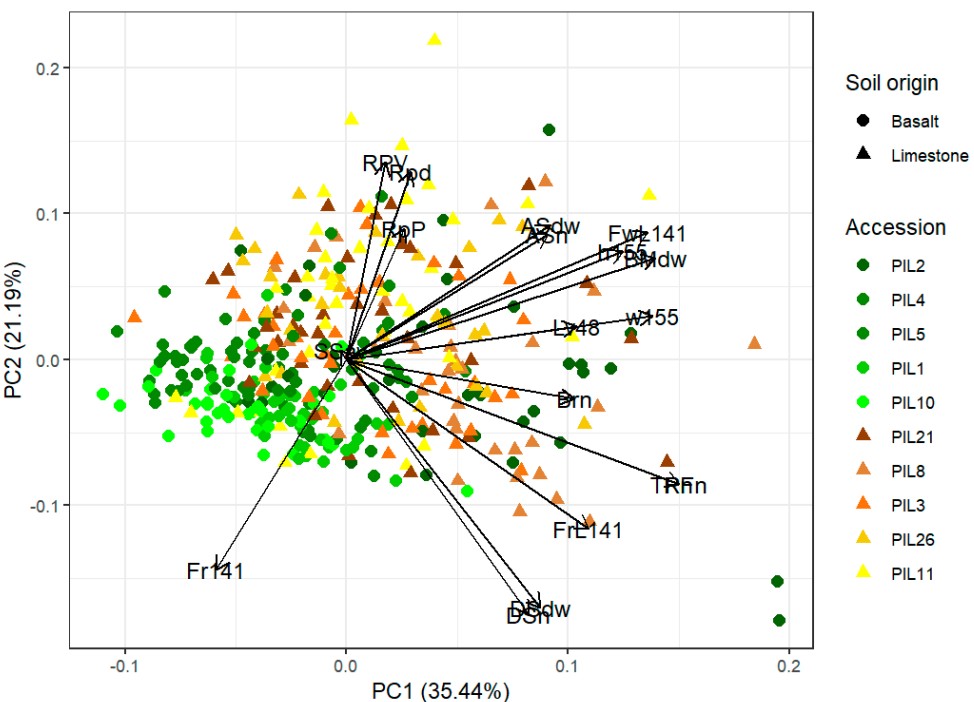

**Figure 3.** Principal component analysis (PCA) of plant traits across accessions. The following traits were included in this analysis: Lv48 = number of leaves per plant (48 days from sowing); Fr141 = number of pods on the main stem (141 days); FwL141 = appearance of lateral flowers (141); FrL141 = appearance of lateral fruits (141); h155 = plant height (155); w155 = plant diameter (155); Rpd = the day a plant completed ripening of all pods; RpP = ripening duration; Brn = number of branches number; ASn = number of abnormal seeds per plant; Pn = number of pods per plant; ASdw = abnormal seed DW; BMdw = vegetative organs DW; TRFn = number of mature pods; DSn = number of normally developed seeds; DSdw = DW of normally developed seeds; SSav = single seed DW average; RPV = reproductive/vegetative DW ratio.

### 3.5. Effect of Honeybee Pollination on L. pilosus Success

Five accessions (1, 2, 3, 4, and 10) did not show a significant effect of honeybee pollination on seed yield (Figure 4). Five accessions showed a significant effect of honeybee pollination on fully developed seed DW, accessions 5, 8, 11, 21, and 26. These accessions are limestone-related and showed a significantly higher yield when exposed to honeybee pollination—91% increase in average, $64 \pm 5$ (SE) g DW seed without pollination, and $123 \pm 10$ (SE) g DW seed with pollination. Only accession 5, of basalt origin, showed a significant reduction in yield when exposed to bee pollination. Exposure to pollination significantly shortened the duration from the onset of pod drying to seed shattering in four accessions, without preserving the pattern of pollination's effect on seed yield or showing any correlation with soil origin. Specifically, the duration decreased by 16% in accession 4, by 16% in accession 5, by 29% in accession 11, and by 21% in accession 21. All other accessions did not exhibit significant changes in this duration when exposed to pollination.

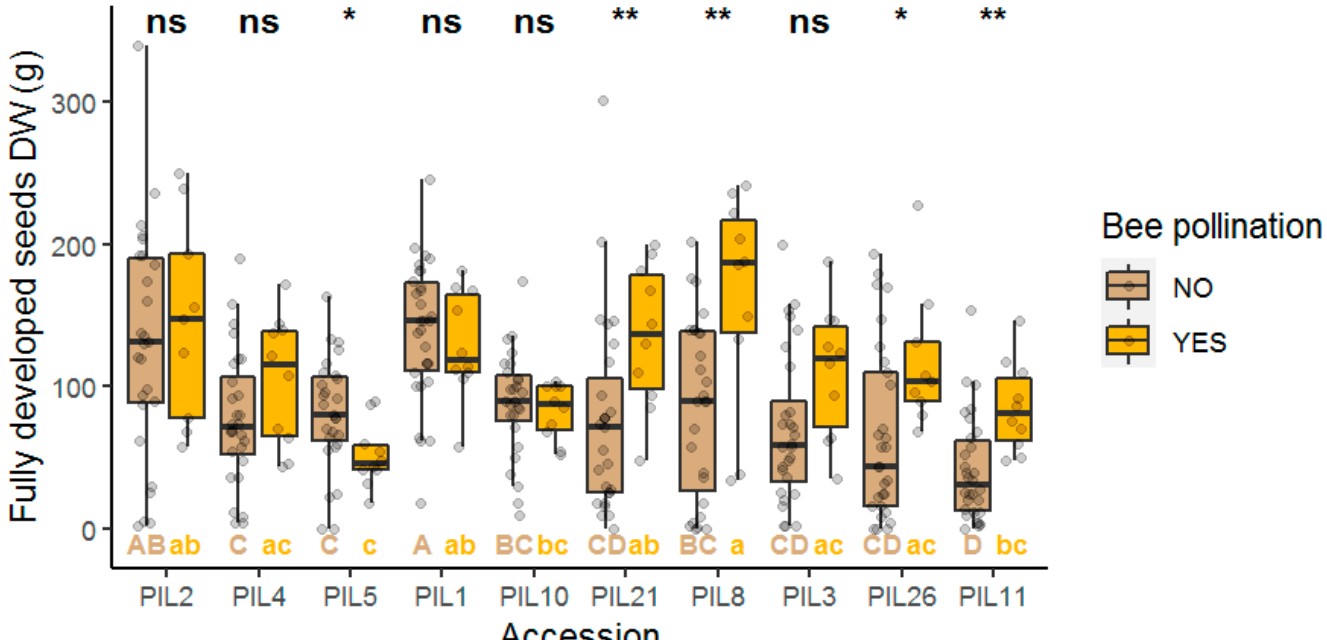

**Figure 4.** Impact of honeybee pollination on yield. DW (g) of fully developed seeds with and without honeybee pollination is represented as an average per plant. Asterisks at the upper bolded line represent significant differences, determined by the Wilcoxon nonparametric test, between pollinated versus unpollinated accession, where ns = non-significant ($p > 0.05$); * = significantly different ($p < 0.05$) ** ($p < 0.01$). Significant differences between self-pollinated accessions are denoted by different capital letters, determined by the Kruskal–Wallis nonparametric test followed by the post hoc Dunn test (Benjamani–Hochberg method). Different small case letters represent significant differences between honeybee pollinated accessions, determined after square root transformation by ANOVA test followed by Tukey HSD post hoc test.

### 3.6. Seed Yield in High Density

We compared seed yield when grown at low and high plant densities, 2.45 and 6.55 plants/m$^2$, respectively. The results do not show an apparent effect of planting density on *L. pilosus* final seed yield per area. Moreover, the first generation of wild lupin seeds produced a yield equivalent to other commercial lupins and soybeans (Figure 5).

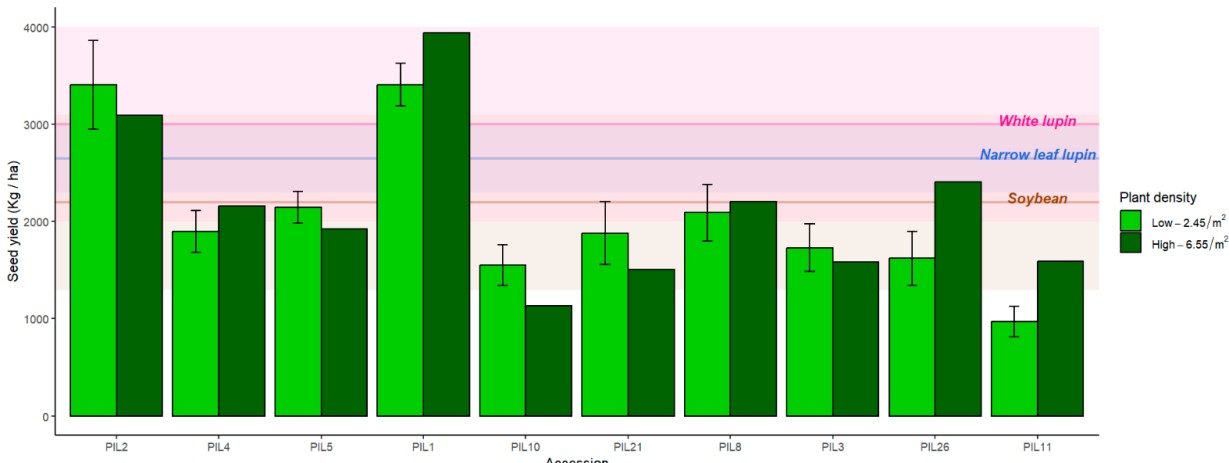

**Figure 5.** Effect of plant density on *L. pilosus* seed yield. Seed yield in kg per hectare and SE (by error bars) for low-density (2.45 plants per m$^2$) versus high-density conditions (6.55 plants per m$^2$). For reference, see range and mean (solid line) (pale background color) and average (darker line) of white lupin (*L. albus*), narrow leaf lupin (*L. angustifolius*), and soybean (*Glycine max*). Data for *L. albus* was acquired from the Horizon2020 project Legume Hub Project, grant number 817634 [60], *L. angustifolius* [61], and soybean [62].

## 4. Discussion

***L. pilosus* as a prospective novel crop—agronomic aspects.** Endeavors to examine wild lupins as a potential crop should consider their nutritional values (primarily their protein content), toxicity (alkaloid profile), and agricultural compatibility. This agricultural value include yield, seed- and pod-shattering avoidance [63], and pathogen and pest resistance. Here, we focused on the most important agricultural aspects. Our results show that wild populations of *L. pilosus* can grow in agricultural field conditions with high success rates, strong evidence for good physiological performance, and a high seed yield. Only 2% of plants did not complete their life cycle, suffering from poor germination or phytopathology. We did not support the plants with fertilization, herbicides, or pesticides. No nutrient deficiencies were observed during the growing season, which correlates with sufficient nutrient levels at the beginning of the season. Despite some pathologies, all lupin populations exhibited successful growth and seed production. The average total yield of seeds was equivalent to 2 tons per hectare, which meets the standard range of relative legume production (Figure 5). The first generation of wild *L. pilosus* in cultivated field conditions produced high quality seeds, with a high proportion of normally developed seeds (Figure 2c).

Currently, the lack of knowledge regarding the nutritional and non-nutritional properties of the yield is fundamental. Nevertheless, considering the high yields per area compared to established pulse crops during our first attempt to grow wild lupins, we conclude that the first hypothesis is supported; *L. pilosus* has potential as a promising novel protein crop. Recently, research suggests that for some uses of the food industry, the high alkaloid content of wild lupins can be overcome by methods for protein isolation and purification [24,64], allowing the use of novel protein crops [65].

**How geographically separated populations of *L. pilosus* differ agronomically**. Our second hypothesis is partially supported—the results showed significant differences between the offspring of geographically separated *L. pilosus* populations. Accessions originating in a limestone environment developed faster and entered the reproductive growth stage several weeks before the accessions of basalt origin (Figure 1). The short vegetative stage of the limestone accessions was followed by a more extended reproductive period with long inflorescence and fruit set, and a slightly shorter ripening time. In addition, the architecture of the limestone accessions appeared different from that of the basalt accessions. Basalt populations grew shorter, exhibiting a cushion-like structure, whereas limestone accessions

grew taller. The different architecture explains the higher reproductive/vegetative tissue ratio in the basalt accessions (Figure 2d). Finally, the basalt accessions produced 34% higher seed DW than the limestone accessions (Figure 2). The overall average biomass of a single *L. pilosus* seed (0.59 ± 0.01 (SE) g DW) was similar to the average biomass of the seeds of the mother plants (Table 2), supporting the theory suggesting that seed size is a conservative component in many crop plants [66]. Interestingly, only limestone accessions showed a significant increase in seed yield when exposed to honeybee pollination—nearly doubling their yield (Figure 4). The basalt accessions did not show an increase in yield with pollination, and one case (accession 5) even experienced a reduction. The different impact of honeybee pollination may be explained by phenological differences between the basalt and limestone accessions—early blooming may respond better to the introduction to honeybee activity. Ultimately, a future lupin crop will grow in an open field and be exposed to foreign pollination by honeybees and other insects. The results suggest that commercialized pollinated limestone accessions would likely overcome the observed advantage of the basalt accessions in seed yield. The honeybee experiment also suggests that the potential yield in open fields is almost twice that of the 2 tons per hectare we observed in the closed net-house chambers.

**Exploring dimensions of wild populations as a strategy for de novo domestication or commercializing biological resources**. The distinguished phenology, structure, and yield differences between basalt and limestone accessions appeared in the PCA (Figure 3). The overlap between accessions suggests that the different accessions of *L. pilosus* are not distinguishably different from each other. However, more broadly, the basalt-origin accessions differ from the limestone accessions. Exploring traits for domestication requires then a more thorough survey of wild accessions, which can ultimately support diverse processes of domestication [67]. Moreover, understanding the potential of *L. pilosus* as a novel protein crop requires the study of the fundamental characteristics of the potential crop that we did not explore in this study. The most important missing information includes the links between environmental conditions in the place of origin and the following plant traits: alkaloid content, nutritional value of the seeds, and avoidance of pod shattering [68]. Nevertheless, for a high-value crop, some of these obstacles can be addressed by modern methods, allowing the separation of proteins from the seeds [24], the reduction in pod shattering [63], or targeting green pod yields rather than dry mature seeds. This study demonstrates the value of linking the ecological background of different populations and agronomic traits of a focal species as a target for future commercial use.

*L. pilosus* **grew in high density**. We performed an initial examination of density-dependent *L. pilosus* development. The results show that the seed yield per plant of all accessions was not significantly reduced by higher-density planting (Figure 5). These results suggest that while growing *L. pilosus* as a crop in an open field, the sowing density can be closer to 6.25 plants/m$^2$ rather than 2.38 plants/m$^2$. Sowing lupins with greater plant densities is more competitive with weeds, soil erosion, and pests, while plant loss for competition is compensated and harvesting is more straightforward. Overall, it is suggested that high lupin density offers "good insurance at a minimal cost" [48].

## 5. Conclusions

This research provides strong evidence for the agronomic compatibility of wild *L. pilosus*, showcasing its potential as a novel crop with a high seed yield and adaptability to diverse environmental conditions. Our findings reveal distinct agronomic traits among geographically separated populations, particularly between basalt and limestone habitats, underscoring the role of ecology in shaping yield and plant characteristics.

To the best of our knowledge, this research is among the first to comprehensively evaluate *L. pilosus*'s potential in an agricultural context of Mediterranean agricultural ecosystems, contributing valuable insights into crop diversification for improved food security. The study's innovative approach highlights how preserving these unique phenotypic traits

can enhance crop resilience and yield stability, making *L. pilosus* a promising candidate for sustainable agriculture.

Our approach of incorporating wild plants into local agricultural operations leverages ecological adaptations to changing climates, capitalizing on natural variation caused by diverse local ecological conditions. By identifying high-producing accessions and integrating them with advanced post-harvest processing techniques, we can effectively mitigate the anti-nutritive secondary metabolites present in these wild plants, facilitating their rapid integration into local cropping systems. Ultimately, the use of wild plants as crops holds potential to diversify food along its value chain.

By evaluating the ways to diversify agricultural production, this type of research could unveil the vast potential of using wild plants as novel crops for more sustainable food production. This strategy not only enhances crop diversity but also contributes to the resilience of locally adapted agricultural systems in the face of climate change and evolving environmental pressures.

**Supplementary Materials:** The following supporting information can be downloaded at: https://www.mdpi.com/article/10.3390/agronomy14122804/s1, Supplementary Material S1: Aerial image of the net-house experiment; Supplementary Material S2: Photos of the growth experiment for examination of *L. pilosus* as a novel crop; Supplementary Material S3: Phenological index of *L. pilosus*.

**Author Contributions:** E.B.-S.—Conceptualization; Data curation; Formal Analysis; Methodology; Software; Visualization; Writing—review and editing. O.S.—Conceptualization; Funding acquisition; Investigation; Project administration; Supervision; Writing—original draft, review and editing. M.S.—Conceptualization; Investigation; Methodology; Resources; Writing—review and editing. O.C.—Conceptualization; Investigation; Methodology; Resources; Supervision; Writing—review and editing. All authors have read and agreed to the published version of the manuscript.

**Funding:** This research received no external funding.

**Data Availability Statement:** The original contributions presented in the study are included in the article/Supplementary Materials, further inquiries can be directed to the corresponding author.

**Acknowledgments:** The authors are grateful to those who contributed laboratory and field work in this research project—Kinneret Rozenblat, Guy Shotland, Smadar Wininger, and Zvika Amitai. We also thank Neta Shiler for providing illustrations for the phenology index. We acknowledge the researchers and research assistants at the ARO who assisted in this research: Elazar Qvinn identified the pest beetle, Omer Frenkel advised pathology, Yiftach Vaknin advised pollination and other agronomic concerns, and Shmuel Galili provided net-house space for the growth experiments.

**Conflicts of Interest:** The authors declare no conflicts of interest.

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
