# Peer review of "Agronomic Estimation of Lupin (Lupinus pilosus L.) as a Prospective Crop"

_agronomy, doi:10.3390/agronomy14122804_

Round 1

Reviewer 1 Report

Comments and Suggestions for Authors

This manuscript is well written, but further improvements needed.

L 79-84. You need to describe something about previous work done at DPIRD in Western Australia regarding the domestication of L. pilosus. It already has all domestication traits available, but they are not combined into a single genotype for cultivation.  This needs to be reported with an updated information.

In your research, you used wild populations and did not mention about whether they already had domesticated traits. If not, how they are going to be cultivated to diversify the list of high protein crops?

Evaluated 10 wild populations collected mainly from basalt and limestone rock areas. Why do you call them lineages rather than accessions?

Table 2. Although you have given location of collection sites, it is almost no use when a reader is not familiar with those locations. So, provide GPS co-ordinates to visualise how far were these sites. Is it an average seed weight of an individual seed or something else. Please make it clear.

Line 131-32. Describe irrigation cycle, it is not clear. Two hours of irrigation seems to be quite high; describe it.  Write in third person, instead of I or we.

Line 157-59. Instead of just giving the days after seeding, provide the growth stages, such as early veg growth, flowering, podding etc.

Line 161. It is not only vegetative development you have described, but also the reproductive stage.  Change the sub-heading to plant growth stages.

Line 195: Seed damage caused by insect is not called infected, change it to infested.

Line 201: what do you mean by vegetative tissue ratio? Is it harvest index? Please describe it.

Fig 2. I don’t see any importance of these figures. These are just normal procedures and different growth stages of pilosus. For readers who is unfamiliar with pilosus, you may leave e with inflorescence, others can be removed.

The same thing applies to Figure 1. You can describe these steps in the text.

Graphs in in Figs 3-7 are fudgy and need to have a better resolution. All these graphs are poorly drawn with very low resolution and should not have been included in the manuscript.

Line 258: it should be from sowing, not from propagation.

Results have been poorly described. You have emphasised explanation by graphs, but they have poor resolution and cannot be read properly.

What was the rational of including honey bees? What is the mode of pollination on L pilosus? Is it self pollinated or cross pollinated?

Discussion:

Describe the accessions based on basalt and limestone rock origin, effect of pollination by honey bees, Is there an info on pilosus pollination?

References:

Looks like you did not check the reference properly. Ref 1 still has the generic formay listed.  

Comments on the Quality of English Language

I am not qualified enough to comment on language.

Author Response

Dear Editor,

Thank you for considering our manuscript for publication at Agronomy. We thank for three anonymous reviewers for critically reading the manuscript, finding mistakes and asking for improvements. We were encouraged to see that generally, the reviewers thought the manuscript meet the requirements for publication in Agronomy after this revision. The following is our specific response for every issue raised by the reviewers.

On behalf of all authors of this manuscript, Oren Shelef.

Reviewer 1

Comment 1: This manuscript is well written, but further improvements needed.

Revision 1: We thank the reviewer for the detailed revision.

Comment 2:  L 79-84. You need to describe something about previous work done at DPIRD in Western Australia regarding the domestication of L. pilosus. It already has all domestication traits available, but they are not combined into a single genotype for cultivation. This needs to be reported with an updated information.

Response 2: We agree - this is an important part that must be included. We've added a thorough description of the DPIRD team contribution.

Comment 3: In your research, you used wild populations and did not mention about whether they already had domesticated traits. If not, how they are going to be cultivated to diversify the list of high protein crops?

Response 3: This research is focusing on evaluating the agronomic potential of L. Pilosus with emphasis on evaluation of seed production. We've clarified that in the text – intro and discussion. Specifically see our response #2 to comment #2 that is clarifying the former efforts done to domesticate this species. Other clarifications are given in the discussion section.

Comment 4: Evaluated 10 wild populations collected mainly from basalt and limestone rock areas. Why do you call them lineages rather than accessions?

Response 4: After considering this comment and similar critique by the other two reviewers, we agree to this good advice and change the phrasing to "accessions" rather than "lineages" through the entire manuscript.

Comment 5: Table 2. Although you have given location of collection sites, it is almost no use when a reader is not familiar with those locations. So, provide GPS co-ordinates to visualise how far were these sites.

Response 5: GPS co-ordinates are now provided within Table 2.

Comment 6: Is it an average seed weight of an individual seed or something else. Please make it clear.

Response 6: seed weight value is now clarified in Table 2, among all other parameters: # = accession number; Collected = the year in which seeds were collected from the wild population, in the early summer (May – June); Site = domestic name of the site location; Soil = basalt or limestone; Altitude = in meters above see level (m a.s.l.); Location = GPS coordinates, latitude (N) and longitude (E); Average Seed DW = an average seed dry weight in grams, given as an average weight of a single seed.

Comment 7: Line 131-32. Describe irrigation cycle, it is not clear. Two hours of irrigation seems to be quite high; describe it. Write in third person, instead of I or we.

Response 7: The following sentence was added: "The irrigation was conducted by drip irrigation pipe, with controlled-emitters along the pipe, located every 30 cm, each emitting 2 liters per hour". Voice was changed to passive third person.

Comment 8:  Line 157-59. Instead of just giving the days after seeding, provide the growth stages, such as early veg growth, flowering, podding etc.

Response 8: We added as requested the following full description of measurement timing: early vegetative growth (59 dyas from seeding), after first inflorescence of the limestone l accessions  (90 dyas from seeding, and after complete fruitsetting of all accessions (155 days from seeding)

Comment 9: Line 161. It is not only vegetative development you have described, but also the reproductive stage.  Change the sub-heading to plant growth stages.

Response 9: Correct. We've changed the sub-heading.

Comment 10: Line 195: Seed damage caused by insect is not called infected, change it to infested.

Response 10: changed to infested.

Comment 11: Line 201: what do you mean by vegetative tissue ratio? Is it harvest index? Please describe it.

Response 11: thank you for this clarification – changed to "reproductive-vegetative tissue ratio"

Comment 12: Fig 2. I don’t see any importance of these figures. These are just normal procedures and different growth stages of pilosus. For readers who is unfamiliar with pilosus, you may leave e with inflorescence, others can be removed.

Response 12: Figures 1-2 were removed to the supplemental materials.

Comment 13: The same thing applies to Figure 1. You can describe these steps in the text

Response 13: Figures 1-2 were removed to the supplemental materials.

Comment 14: Graphs in in Figs 3-7 are fudgy and need to have a better resolution. All these graphs are poorly drawn with very low resolution and should not have been included in the manuscript.

Response 14: All figures were uploaded separately in good resolution (300 dpi, TIFF files with over 1M resolution. If the embedded figures are not in the best possible resolution, this seems to be a technical issue that the journal can direct authors how to improve.

Comment 15: Line 258: it should be from sowing, not from propagation.

Response 15: changed to "sowing" accordingly.

Comment 16: Results have been poorly described. You have emphasised explanation by graphs, but they have poor resolution and cannot be read properly.

Response 16: We now elaborated the description of results and discussion. See comment 14.

Comment 17: What was the rational of including honey bees? What is the mode of pollination on L pilosus? Is it self pollinated or cross pollinated?

Response 17: While L. pilosus is primarily self-pollinated, honeybee pollination was introduced to potentially boost seed yield and uniformity, as cross-pollination can occasionally enhance seed production in legumes. This was clarified in the intro and discussion.

Comment 18: Describe the accessions based on basalt and limestone rock origin, effect of pollination by honey bees, Is there an info on pilosus pollination?

Response 18: See response to comment 17.

Comment 19: Looks like you did not check the reference properly. Ref 1 still has the generic formay listed. 

Response 19: All references were re-edited according to the journal standards.

We would like to thank again the reviewers and editors for providing us with the opportunity to improve our manuscript.

Sincerely,

Oren Shelef, PhD

Researcher in Agroecology

Agricultural Research Organization - Volcani Institute, Israel

+972-526610931 | Shelef@volcani.agri.gov.il

Reviewer 2 Report

Comments and Suggestions for Authors

Row 2 (in Title):

Authors wrote: "Agronomic estimation of Lupinus pilosus as a novel protein crop"

Reviewer: It would be more easier for read if will Authors write the native name of species in english language in front/behind the Lupinus pilosus.

Reviewer, maybe: "Agronomic estimation of wild lupin (Lupinus pilosus) as a novel protein crop"

Please correct Title.

Row 2 (in Title):

Authors wrote: "Agronomic estimation of Lupinus pilosus as a novel protein crop"

Reviewer: In the Title Authors wrote: "novel protein crop".

My question to Authors: How do you know that wild populations of Lupinus pilosus in Your completed experiment have protein content? Nowhere in completed experiment Authors did not analysed protein content in ten wild lupin populations. Authors analysed only agronomic traits, not quality content in wild lupin.

Reviewer: maybe change words "novel protein crop " to "possible novel protein crop" or some other agronomical word that Authors investigate in their experiment.

"Novel protein crop " to "possible novel protein crop" must be correct in complete Manuscript.

Please explain and correct in Manuscript.

Row 30 (in 1. Introdustion):

Authors wrote: Authors divided Introduction to two parts. First part is about: 1.1. Sustainable food production and novel protein-rich crops. Second part is about: 1.2. Lupins as a protein crop.

Reviewer: First part of the Introduction should be dropped. Authors in this part dont give direct answer to "Agronomic estimation of Lupinus pilosus as a novel protein crop".

Second part of Introduction is targeting the title of Manuscript. In this part Authors define four aims of this study. Also, Introduction contain 33 different literature citations. From this 33 literature citations only five (5) are directly targeting "Lupinus pilosus". 28 literature citations are describes idea of manuscript on the different point of view. For better describing actual problems in this field of research Authors must make use of more recent literature sources with similar investigations on the Lupinus pilosus as a novel protein crop. Maybe it will be useful to write couple sentences/citations that are close to four aims of this study that Authors wrote in the second part of Introduction. From this reasons, Introduction must be rewrite again with targeting literature sources and citations to the main goals of investigation on the Lupinus pilosus.

Please dropped first part of Introduction and rewrite the rest of Introduction.

Row 84 (in 1. Introdustion):

Authors wrote:produces large seeds in natural populations (~550mg); 2) L. pilosus...

Reviewer: this quantity of seed (~550mg) is per plant, per area, or per what ???

Please describe and correct sentence.

Row 138 (in 2. Materials and Methods):

Authors wrote: … Olsen extraction method (Olsen et al., 1954) followed...

Reviewer:Wrong type of citation. Citations must be given with numbers, not by names.

Please correct citation.

Row 176 (in 2. Materials and Methods):

Authors wrote: … about 80% or 0.8 Fv/Fm (Shelef et al., 2011; 2019). We measured...

Reviewer:Wrong type of citation. Citations must be given with numbers, not by names.

Please correct citation.

Row 182 (in 2. Materials and Methods):

Authors wrote: … 5 – yellow, chlorotic (Brand et al., 1999). Throughout...

Reviewer:Wrong type of citation. Citations must be given with numbers, not by names.

Please correct citation.

Row 203 (in 2. Materials and Methods):

Authors wrote: 2.5. Data analysis

Reviewer: in row 185 Authors wrote: 2.5. Reproductive growth parameters.

How can two diffrent subchapters in same Chapter have the same number?

Please explain and correct number of subchapters.

Rows 257, 274, 283, 294, 306, 314, (in 3.Results):

Authors wrote: numbers of subchapters: 3.1.1.; 3.1.2.; 3.1.3.;3.1.4.; 3.1.5.; 3.1.6.;

Reviewer:if number of Chapter Results is "3." , than subchapters in this chapter must have numbers: 3.1 ; 3.2 ; 3.3 ; 3.4; 3.5 ; 3.6 .

Please correct numbers of subchapters.

Rows from 274 to 282 (in 3.Results):

Authors wrote: In this subchapter Authors use different types of abbreviations for yield : (SE) g DW; (SE); g DW; (SE); seeds (SE) per plant; g (SE) DW; (se); (se) g DW; (se) g DW per seed; (se) g DW per seed.

Also Authors use two type of similar abbreviations: "se" and "SE". Are they same or not?

Reviewer: please explain this abbreviations somewhere.

Rows 330 (Table 1. in 3.Results):

Authors wrote: Table 1. List of the examined Domestication Agronomic Traits.

Reviewer: In Table 1. is not clear for reader which trait belong to which Type. Give some line separations or empty row between two different types, to see where the new trait is starting.

Please correct this in Table 1.

Rows 332 (in 3.Results):

Authors wrote: The text continues here (Figure 2 and Table 2).

Reviewer: What is the meaning of this sentence?

Please explain and correct.

Rows 345 (Figure 4. in 3.Results):

Authors wrote: In Figure 4. part: a,b,d -Authors wrote word: "basalt"

Reviewer: maybe: basal ?

Please explain and correct this word.

Rows 379 (in Figure 6.):

Authors wrote: …significantly different (p < 0.05) ** (p < 0.01).

Reviewer: In rows 225 (in 2. Materials and Methods) Authors wrote: we considered differences as statistically significant at p=0.05 .

If Authors use this level of significance (0.01) in the Manuscript, than must it be also wrote in the Materials and Methods. In Materials and methods Authors stated only 0.05.

Please explain and correct level of significance (0.01) in the Manuscript.

Rows 388-390 (in 3. Results, Figure 7.):

Authors wrote: Data for L. albus was acquired from the Horizon2020 project (Arncken et al., 2020), L. angustifolius by Panasiewicz et al. (2020), and soybean by the FAO (2021).

Reviewer:Wrong type of citations. Citations must be given with numbers, not by names.

Please correct citations.

Rows 401 (in 4. Discussion):

Authors wrote:... potassium levels were slightly high...

Reviewer: How you know that was slightly high?

Explain please potassium levels.

Rows 443 and 445 (in 4. Discussion):

Authors wrote:... basalt...

Reviewer: maybe basal ?

Please explain and correct this word.

Rows 391 to 464 (in 4. Discussion):

Authors used seven different citations in four separate unit of Discussion: 21, 34, 40, 41, 42, 43, 44,

Reviewer: this is very small quantity of citations. Every separate unit in Discussion must have minimum three to four different citations targeting Agronomical aspects of wild lupins. Not other crops or species.

Please complete Discussion with new citations.

Rows 461 to 464 (in 4. Discussion):

Authors wrote: Gladstone (1998) stresses that sowing lupins with greater plant densities is more competitive with weeds, soil erosion, and pests, while plant loss for competition is compensated and harvesting is more straightforward. Gladstone suggests high lupin density is "good insurance at a minimal cost."

Reviewer: Authors wrote wrong type of citation and use same citation twice in two connected sentences.

Reviewer, maybe write like this: Sowing lupins with greater plant densities is more competitive with weeds, soil erosion, and pests, while plant loss for competition is compensated and harvesting is more straightforward. High lupin density is "good insurance at a minimal cost" (34).

Please correct this two sentences with same citation.

Rows from 465 to 476 (in 5.Conclusions):

Authors wrote: Conclusions are full of citations (five time) without Authors results and answers to the Title of Manuscript.

Reviewer: Are there in this Conclusions any Authors direct Conclusion??? If Authors completed the experiment with wild lupins, than must have their own conclusions. In Conclusions Authors use citations (five time) (20, 45, 46, 47, 47) and which were written in this Chapter.

Reviewer: Very important Chapter of Manuscript is "Conclusion" and in this Manuscript is Very simple and short. Conclusions must contain direct Authors findings from their research and not only general notes. Conclusions must be consistent with the evidence and arguments presented in Results and Discussion. In Conclusions write the specific Agronomic estimation of Lupinus pilosus as a novel protein crop that are suggest in Manuscript. This Chapter must be rewrite again according the Journal Guide for Authors. Conclusions must be rewrite again based on Authors Results from completed experiment and Discussion without any citation.

Please rewrite and correct Conclusions.

Rows from 496 to 583 (in References):

Authors wrote: 47 different literature sources (First reference is not "reference").

Reviewer: Authors wrote" 48 References" not acording the Journal Guide for Authors.

Every single reference in the List of References has some error or imperfection.

Authors names are not written correctly;

Authors use words that are not allowed between Authors in this Chapter like: "and"; "et.al" ;

All Journal Names must be written in "Italic" mode; they are not now,

Year of publication must be written in "bold" mode; they are not now,

Page range must be written without abbreviation: "p.";

This Chapter must be rewrite again according the Journal Guide for Authors.

Please rewrite and correct References.

Author Response

Dear Editor,

Thank you for considering our manuscript for publication at Agronomy. We thank for three anonymous reviewers for critically reading the manuscript, finding mistakes and asking for improvements. We were encouraged to see that generally, the reviewers thought the manuscript meet the requirements for publication in Agronomy after this revision. The following is our specific response for every issue raised by the reviewers.

On behalf of all authors of this manuscript, Oren Shelef.

Reviewer 2

Comment 1: Authors wrote: "Agronomic estimation of Lupinus pilosus as a novel protein crop". It would be more easier for read if will Authors write the native name of species in english language in front/behind the Lupinus pilosus. Reviewer, maybe: "Agronomic estimation of wild lupin (Lupinus pilosus) as a novel protein crop". Please correct Title.

Response 1: The title was changed as advised.

Comment 2: Authors wrote: "Agronomic estimation of Lupinus pilosus as a novel protein crop". Reviewer: In the Title Authors wrote: "novel protein crop".

My question to Authors: How do you know that wild populations of Lupinus pilosus in Your completed experiment have protein content? Nowhere in completed experiment Authors did not analysed protein content in ten wild lupin populations. Authors analysed only agronomic traits, not quality content in wild lupin... maybe change words "novel protein crop " to "possible novel protein crop" or some other agronomical word that Authors investigate in their experiment. "Novel protein crop" to "possible novel protein crop" must be correct in complete Manuscript. Please explain and correct in Manuscript.

Response 2: After deleting the word "protein" as the reviewer suggested, the title reads now as: "Agronomic evaluation of Lupin (Lupinus pilosus) as a novel protein crop". The focus of this research is not in the nutritional values of L. pilosus seeds – but rather in the potential yield production, based on 10 accessions with distinct differences in their local environments. This focused objective is now clarified within the text.

Comment 3: Row 30 (in 1. Introdustion): Authors wrote: Authors divided Introduction to two parts. First part is about: 1.1. Sustainable food production and novel protein-rich crops. Second part is about: 1.2. Lupins as a protein crop. First part of the Introduction should be dropped. Authors in this part dont give direct answer to "Agronomic estimation of Lupinus pilosus as a novel protein crop".

Response 3: The introduction is meant to give a wide background to the research topic, starting from sustainable food chain, to plant-based protein production, legumes and lupins, then lupins as crop, and specifically L. pilosus. We think this flow bring the reader to the research with a wide scope and valuable context. We re-edited the entire introduction to make sure it's more focused on the necessary background to our research goals and focus.

Comment 4: Second part of Introduction is targeting the title of Manuscript. In this part Authors define four aims of this study. Also, Introduction contain 33 different literature citations. From this 33 literature citations only five (5) are directly targeting "Lupinus pilosus". 28 literature citations are describes idea of manuscript on the different point of view. For better describing actual problems in this field of research Authors must make use of more recent literature sources with similar investigations on the Lupinus pilosus as a novel protein crop. Maybe it will be useful to write couple sentences/citations that are close to four aims of this study that Authors wrote in the second part of Introduction. From this reasons, Introduction must be rewrite again with targeting literature sources and citations to the main goals of investigation on the Lupinus pilosus. Please dropped first part of Introduction and rewrite the rest of Introduction.

Response 4: This study builds on previous research on L. pilosus as a high-potential legume crop (e.g., Heistinger & Pistrick, 2007; Nelson & Hawthorne, 2000). In refining our introduction, we focused on literature most relevant to the agronomic traits of L. pilosus to underscore its potential in sustainable agriculture

Finally, as explained in Response # 3, the manuscript is re-edited to link between the first part of our introduction giving a wider scope of sustainable agriculture and the potential of using local wild plants as novel crops. More relevant references were added.

Heistinger, A., & Pistrick, K. (2007). ‘Altreier Kaffee’: Lupinus pilosus L. cultivated as coffee substitute in northern Italy (Alto Adige/Südtirol). Genetic resources and crop evolution54, 1623-1630.

Nelson, P., & Hawthorne, W. A. (2000, May). Development of lupins as a crop in Australia. In Linking Research and Marketing Opportunities for Pulses in the 21st Century: Proceedings of the Third International Food Legumes Research Conference (pp. 549-559). Dordrecht: Springer Netherlands.

Comment 5: Row 84 (in 1. Introdustion): Authors wrote: …produces large seeds in natural populations (~550mg); 2) L. pilosus... this quantity of seed (~550mg) is per plant, per area, or per what ??? Please describe and correct sentence.

Response 5: We added the clarifying words "~550mg per a single seed"

Comment 6: Row 138 (in 2. Materials and Methods): Authors wrote: … Olsen extraction method (Olsen et al., 1954) followed...Wrong type of citation. Citations must be given with numbers, not by names. Please correct citation.

Response 6: All references were re-edited.

Comment 7: Row 176 (in 2. Materials and Methods) Authors wrote: … about 80% or 0.8 Fv/Fm (Shelef et al., 2011; 2019). We measured... Wrong type of citation. Citations must be given with numbers, not by names. Please correct citation.

Response 7: All references were re-edited.

Comment 8: Row 182 (in 2. Materials and Methods) Authors wrote: … 5 – yellow, chlorotic (Brand et al., 1999). Throughout... Wrong type of citation. Citations must be given with numbers, not by names. Please correct citation.

Response 8: All references were re-edited.

Comment 9: Row 203 (in 2. Materials and Methods) Authors wrote: 2.5. Data analysis. in row 185 Authors wrote: 2.5. Reproductive growth parameters. How can two diffrent subchapters in same Chapter have the same number? Please explain and correct number of subchapters.

Response 9: Subsection numbers were re-edited.

Comment 10: Rows 257, 274, 283, 294, 306, 314, (in 3.Results). Authors wrote: numbers of subchapters: 3.1.1.; 3.1.2.; 3.1.3.;3.1.4.; 3.1.5.; 3.1.6.; if number of Chapter Results is "3.", than subchapters in this chapter must have numbers: 3.1 ; 3.2 ; 3.3 ; 3.4; 3.5 ; 3.6. Please correct numbers of subchapters.

Response 10: Subsection numbers were re-edited.

Comment 11: Rows from 274 to 282 (in 3. Results). Authors wrote: In this subchapter Authors use different types of abbreviations for yield : (SE) g DW; (SE); g DW; (SE); seeds (SE) per plant; g (SE) DW; (se); (se) g DW; (se) g DW per seed; (se) g DW per seed. Also Authors use two type of similar abbreviations: "se" and "SE". Are they same or not? Reviewer: please explain this abbreviations somewhere.

Response 11:

Comment 12: Rows 330 (Table 1. in 3.Results). Authors wrote: Table 1. List of the examined Domestication Agronomic Traits. Reviewer: In Table 1. is not clear for reader which trait belong to which Type. Give some line separations or empty row between two different types, to see where the new trait is starting. Please correct this in Table 1.

Response 12: We tried to follow the journal layout – which seems to be without separation lines. After this comment we added separation lines between "Type" sections for clarification.

Comment 13: Rows 332 (in 3. Results): Authors wrote: The text continues here (Figure 2 and Table 2). Reviewer: What is the meaning of this sentence? Please explain and correct.

Response 13: This is a typo created when we tried to fit our text into the journal format, and missed a guiding sentence. Deleted

Comment 14: Rows 345 (Figure 4. in 3.Results). Authors wrote: In Figure 4. part: a,b,d -Authors wrote word: "basalt" Reviewer: maybe: basal? Please explain and correct this word.

Response 14: Basalt is a soil type formed from the erosion of basalt-rock, a formation made by rapid cooling of lava. This is explained in the text and the comparison between basalt-origin accessions compared to limestone-origin accessions. Basalt is mentioned in the text 34 times. We don't understand why only the "basalt" in figure 4 is questioned. 

Comment 15: Rows 379 (in Figure 6.). Authors wrote: …significantly different (p < 0.05) ** (p < 0.01). Reviewer: In rows 225 (in 2. Materials and Methods) Authors wrote: we considered differences as statistically significant at p=0.05. If Authors use this level of significance (0.01) in the Manuscript, than must it be also wrote in the Materials and Methods. In Materials and methods Authors stated only 0.05. Please explain and correct level of significance (0.01) in the Manuscript.

Response 15: We've addressed this issue shortly with additional text in the materials and methods at the end of section 2.7.2.

Comment 16: Rows 388-390 (in 3. Results, Figure 7.): Authors wrote: Data for L. albus was acquired from the Horizon2020 project (Arncken et al., 2020), L. angustifolius by Panasiewicz et al. (2020), and soybean by the FAO (2021). Reviewer:Wrong type of citations. Citations must be given with numbers, not by names. Please correct citations.

Response 16: All references were re-edited.

Comment 17: Rows 401 (in 4. Discussion): Authors wrote:... potassium levels were slightly high... Reviewer: How you know that was slightly high? Explain please potassium levels.

Response 17: We evaluated NPK levels by soil analysis, before sowing. However, since this analysis was not performed with sufficient repetisions to report a robust evaluatio nof the soil chemistry – we accept the comment and deleted the sentence.

Comment 18: Rows 443 and 445 (in 4. Discussion): Authors wrote:... basalt... Reviewer: maybe basal ? Please explain and correct this word.

Response 18: see Response # 14.

Comment 19: Rows 391 to 464 (in 4. Discussion): Authors used seven different citations in four separate unit of Discussion: 21, 34, 40, 41, 42, 43, 44. Reviewer: this is very small quantity of citations. Every separate unit in Discussion must have minimum three to four different citations targeting Agronomical aspects of wild lupins. Not other crops or species. Please complete Discussion with new citations.

Response 19: We re-edited the discussion and conclusion sections, including the references within.

Comment 20: Rows 461 to 464 (in 4. Discussion) Authors wrote: Gladstone (1998) stresses that sowing lupins with greater plant densities is more competitive with weeds, soil erosion, and pests, while plant loss for competition is compensated and harvesting is more straightforward. Gladstone suggests high lupin density is "good insurance at a minimal cost." Reviewer: Authors wrote wrong type of citation and use same citation twice in two connected sentences. Maybe write like this: Sowing lupins with greater plant densities is more competitive with weeds, soil erosion, and pests, while plant loss for competition is compensated and harvesting is more straightforward. High lupin density is "good insurance at a minimal cost" (34). Please correct this two sentences with same citation.

Response 20: All references were re-edited.

Comment 21: Rows from 465 to 476 (in 5. Conclusions). Authors wrote: Conclusions are full of citations (five time) without Authors results and answers to the Title of Manuscript. Reviewer: Are there in this Conclusions any Authors direct Conclusion??? If Authors completed the experiment with wild lupins, than must have their own conclusions. In Conclusions Authors use citations (five time) (20, 45, 46, 47, 47) and which were written in this Chapter. Reviewer: Very important Chapter of Manuscript is "Conclusion" and in this Manuscript is Very simple and short. Conclusions must contain direct Authors findings from their research and not only general notes. Conclusions must be consistent with the evidence and arguments presented in Results and Discussion. In Conclusions write the specific Agronomic estimation of Lupinus pilosus as a novel protein crop that are suggest in Manuscript. This Chapter must be rewrite again according the Journal Guide for Authors. Conclusions must be rewrite again based on Authors Results from completed experiment and Discussion without any citation. Please rewrite and correct Conclusions.

Response 21: We re-edited the discussion and conclusion sections, including the references within. However, note that the current guideline for editors is rather related to "Free Format Submission" and the journal states that "we do not have strict formatting requirements"

Comment 22: Rows from 496 to 583 (in References) Authors wrote: 47 different literature sources (First reference is not "reference"); Authors wrote" 48 References" not acording the Journal Guide for Authors; Every single reference in the List of References has some error or imperfection.

Authors names are not written correctly; Authors use words that are not allowed between Authors in this Chapter like: "and"; "et.al"; All Journal Names must be written in "Italic" mode; they are not now, Year of publication must be written in "bold" mode; they are not now; Page range must be written without abbreviation: "p."; This Chapter must be rewrite again according the Journal Guide for Authors. Please rewrite and correct References.

Response 22: Note that the guidelines to authors suggests that "Your references may be in any style, provided that you use the consistent formatting throughout". However, all references were re-edited, including the first reference which is a typo, left from miss-patching the reference list within the journal template.

We would like to thank again the reviewers and editors for providing us with the opportunity to improve our manuscript.

Sincerely,

Oren Shelef, PhD

Researcher in Agroecology

Agricultural Research Organization - Volcani Institute, Israel

+972-526610931 | Shelef@volcani.agri.gov.il

Reviewer 3 Report

Comments and Suggestions for Authors

Manuscript: Agronomy 3269525

Agronomic estimation of Lupinus pilosus as a novel protein crop.   Shrlef et al

The topic of this study is interesting, but I suggest a major revision of the submitted version before the acceptance.

 In the text, the authors use the word lineage to describe the samples. I do not agree with this choice, as it is unusual in scientific papers. Moreover, they studied wild populations, which are certainly not made up of genetically uniform seeds, so the word population is more appropriate. Several comments of experimental data are generic definitions (i.e. lines 267 etc). statistical significance of differences, if present, should be reported. Standard deviation should be added in Table 2.

Paragraph 3.1.6 - the comparison with data reported in other studies has a very little significance. Why did the authors not include in the experimental trials at least one commercial cv of lupine?

Line 405-407 The high quality of seeds has not been demonstrated at all. No data on seed composition has been reported. This sentence should be deleted or reworded on the basis of the agricultural traits studied.

The alkaloid issue cannot be solved through protein isolation (lines 412-414).

 Lines 36-38 The loss of biodiversity strongly reduces the possibility to develop new cultivars.

Line 63 technical methods determine the nutritional value of seeds not examine the nutritional value

Line 69 add the author to the scientific name of lupin species

Line 74 I think that also the yield is very important

Line 109 where is cited the Table 1?

Line 112 How many plants for population were sampled?

Line 138 reference cited not according the journal instruction

Line 287 typically developed seeds ???

There is a big confusion in the Reference section. The first reference should be deleted and the number of all cited literature revised.

The ref 4 should be substituted with a more recent study. There are several papers on this topic.

Author Response

Dear Editor,

Thank you for considering our manuscript for publication at Agronomy. We thank for three anonymous reviewers for critically reading the manuscript, finding mistakes and asking for improvements. We were encouraged to see that generally, the reviewers thought the manuscript meet the requirements for publication in Agronomy after this revision. The following is our specific response for every issue raised by the reviewers.

On behalf of all authors of this manuscript, Oren Shelef.

Reviewer 3

Comment 1: The topic of this study is interesting, but I suggest a major revision of the submitted version before the acceptance.

Revision 1: Thank you for reading the manuscript and providing critiques to improve it.

Comment 2:  In the text, the authors use the word lineage to describe the samples. I do not agree with this choice, as it is unusual in scientific papers. Moreover, they studied wild populations, which are certainly not made up of genetically uniform seeds, so the word population is more appropriate.

Response 2: To maintain consistency, we replaced all occurrences of ‘lineage’ with ‘accession’ throughout the manuscript, as per the reviewers’ suggestions

Comment 3: Several comments of experimental data are generic definitions (i.e. lines 267 etc).

Response 3: We read again the M&M section, clarified and explained unclear parts. Specifically, line 267 and on read "The short vegetative stage of the limestone lineages was followed by a more extended period of inflorescence and fruit setting and, finally, a slightly shorter ripening time than the basalt lineages". We now added the relevant numbers of each period.

Comment 4: statistical significance of differences, if present, should be reported. Standard deviation should be added in Table 2.

Response 4: Table 2 is presenting a description of accession origin and details of plant population. There is no meaning for standard deviation of a single population taken in each location. We re-edited the description of accession collection to clarify this.

Comment 5: Paragraph 3.1.6 - the comparison with data reported in other studies has a very little significance. Why did the authors not include in the experimental trials at least one commercial cv of lupine?

Response 5: Paragraph 3.1.6 is read: "We compared seed yield when grown at low and high plant densities, 2.45 and 6.55 plants/m2, respectively. The results do not show an apparent effect of planting density on L. pilosus final seed yield per area. Moreover, the first generation of wild lupin seeds produced a yield equivalent to other commercial lupins and soybeans (Fig. 5)." In this research we did not study several levels of crop density, and just added a single condition of high density. We decided to report these results that are secondary to the major results of seed production.

Including a commercial lupin cv such as L. albus sweet lupin, could indeed expand the insights of this research and provide a stronger comparison of L. pilosus production to that of the existing commercial lupins. However, we decided to rely on literature to estimate the production of the commercial crops, in order to focus our research in the chosen objectives. We added a clarification regarding this aspect – in the manuscript.

Comment 6: Line 405-407 The high quality of seeds has not been demonstrated at all. No data on seed composition has been reported. This sentence should be deleted or reworded on the basis of the agricultural traits studied.

Response 6: The sentence: "Moreover, the first generation of wild L. pilosus in cultivated field conditions produced high quality seeds, as evidenced a high proportion of typically developed seeds". Is rewritten to "The first generation of wild L. pilosus in cultivated field conditions produced a high proportion of normally-developed seeds"

Comment 7: The alkaloid issue cannot be solved through protein isolation (lines 412-414).

Response 7: Methods such as solvent extraction have shown promise in reducing alkaloid content for industrial protein extraction, supporting the potential of L. pilosus as a crop despite naturally high alkaloid levels (Lo et al., 2020). The relevant sentence is rephrased now.

Comment 8: Lines 36-38 The loss of biodiversity strongly reduces the possibility to develop new cultivars.

Response 8: This is agreed, but not the focus of our research. Moreover, reviewers 1-2 suggested shortening or even excluding this part of wide scope background regarding diversity in agriculture. Therefore, we think this aspect is not within the scope of this manuscript.

Comment 9: Line 63 technical methods determine the nutritional value of seeds not examine the nutritional value.

Response 9: we changed "…examine nutritional value" to "…determine nutritional value"

Comment 10: Line 69 add the author to the scientific name of lupin species.

Response 10: We've added "L." to L. albus L., L. angustifolius L., L. luteus L. and L. pilosus L. – this addition is quite exceptional. We did not see this in most of the published literature. We are not sure what's the meaning of this L. (Carl Linnaeus?) – so we leave the final decision here for the journal editors.

Comment 11: Line 74 I think that also the yield is very important.

Response 11: Agreed. We added seed yield.

Comment 12: Line 109 where is cited the Table 1?

Response 12: Table 1 is mentioned before, in line 88: "Table 1 summarizes the agronomic traits that we covered for this research".

Comment 13: Line 112 How many plants for population were sampled?

Response 13: We already stated that "Approximately 100 individuals were sampled at each site". At least 350 pods were collected from each population to represent the natural variety of individuals within the population. This is clarified now.

Comment 14: Line 138 reference cited not according the journal instruction

Response 14: All references were re-edited.

Comment 15: Line 287 typically developed seeds ???

Response 15: changed to "normally developed seeds"

Comment 16: There is a big confusion in the Reference section. The first reference should be deleted and the number of all cited literature revised.

Response 16: All references were re-edited.

Comment 17: The ref 4 should be substituted with a more recent study. There are several papers on this topic.

Response 17: All references were re-edited.

We would like to thank again the reviewers and editors for providing us with the opportunity to improve our manuscript.

Sincerely,

Oren Shelef, PhD

Researcher in Agroecology

Agricultural Research Organization - Volcani Institute, Israel

+972-526610931 | Shelef@volcani.agri.gov.il

Round 2

Reviewer 1 Report

Comments and Suggestions for Authors

The authors forgot to rename the figures. Please rename them correctly.

Do not fully agree with this statement at line 426, ‘Any attempt to examine wild lupins as a potential crop should consider their nutritional values (primarily their protein content), toxicity (alkaloid profile), and agricultural compatibility’.

I would say for any plant species to be cultivated as a crop, it must have permeable seed coat and non-shattering seeds and pods. Other traits, such as protein content, alkaloid content, phenology, loss of seed dormancy etc come later as agronomic traits. This needs to be stated as the authors are examining the potential of domestication of L. pilosus.

Author Response

Dear Editor,

We thank reviewer 1 for accepting our revision with minor comments. The following is our specific response to the issues raised by reviewer 1.

On behalf of all authors of this manuscript, thank you for your revision. Oren Shelef.

Comment 1: The authors forgot to rename the figures. Please rename them correctly.

Revision 1: Figure numbers were revised.

Comment 2: Do not fully agree with this statement at line 426, ‘Any attempt to examine wild lupins as a potential crop should consider their nutritional values (primarily their protein content), toxicity (alkaloid profile), and agricultural compatibility’.

I would say for any plant species to be cultivated as a crop, it must have permeable seed coat and non-shattering seeds and pods. Other traits, such as protein content, alkaloid content, phenology, loss of seed dormancy etc come later as agronomic traits. This needs to be stated as the authors are examining the potential of domestication of L. pilosus.

Revision 2: We agree that non-shattering seeds and pods is a fundamental trait for crops, and added this to our text. Notably, an interesting strategy to bypass seed shattering, is by early harvesting of green pods. We mentioned that in the discussion. In our research team we began to study the development of green lupin seeds and specifically the accumulation of Quinolizidine alkaloids in the green seeds. We have some evidence and references to show that green seeds are not as bitter as mature seeds, and in this condition seed shattering is avoided. The current manuscript is not focusing on these aspects.

We thank reviewer 1 for providing us with the opportunity to improve our manuscript.

Respectfully,

Oren Shelef, PhD

Researcher in Agroecology

Agricultural Research Organization - Volcani Institute, Israel

+972-526610931 | Shelef@volcani.agri.gov.il

Reviewer 2 Report

Comments and Suggestions for Authors

Dear Authors

I would like to thanks to Authors on the acceptance of my Review Recommendations for correction of their Manuscript.

Before we close this review and give the agreement for publishing, Authors need to do one correction.

Anyone of the Conclusion is not chapter for any discussion. Conclusions in any manuscript consist only with direct Authors findings.

Discussion is place where Authors can use many citations for comparation their results with other results.

Because of this fact, Authors must delete citation numbers from Conclusions:

Row 503: citation number [65];

Row 515: citation numbers [26,66];

Row 517: citation number [21,67,68];

Row 518: citation number [68].

Author Response

Agronomy_3269525 Revision2: Agronomic estimation of Lupinus pilosus as a novel protein crop.

Dear Editor,

We thank reviewer 2 for accepting our revision with minor comments. The following is our specific response to the issues raised by reviewer 2.

On behalf of all authors of this manuscript, thank you for your constructive comments. Oren Shelef.

Comment 1: I would like to thanks to Authors on the acceptance of my Review Recommendations for correction of their Manuscript.

Before we close this review and give the agreement for publishing, Authors need to do one correction.

Anyone of the Conclusion is not chapter for any discussion. Conclusions in any manuscript consist only with direct Authors findings.

Discussion is place where Authors can use many citations for comparation their results with other results.

Because of this fact, Authors must delete citation numbers from Conclusions:

Row 503: citation number [65];

Row 515: citation numbers [26,66];

Row 517: citation number [21,67,68];

Row 518: citation number [68].

Revision 1: We've revised the discussion and conclusion section accordingly – removing all reference from the conclusion section.

We thank reviewer 2 for providing us with the opportunity to improve our manuscript.

Respectfully,

Oren Shelef, PhD

Researcher in Agroecology

Agricultural Research Organization - Volcani Institute, Israel

+972-526610931 | Shelef@volcani.agri.gov.il